# The Research on Multi-Material 3D Vascularized Network Integrated Printing Technology

**DOI:** 10.3390/mi11030237

**Published:** 2020-02-25

**Authors:** Shuai Yang, Hao Tang, Chunmei Feng, Jianping Shi, Jiquan Yang

**Affiliations:** 1School of Electrical and Automation Engineering, Nanjing Normal University, Nanjing 210023, China; shuaiyang96@163.com (S.Y.); tang.hao96@126.com (H.T.); 61013@njnu.edu.cn (C.F.); 2Jiangsu Key Laboratory of 3D Printing Equipment and Manufacturing, Nanjing Normal University, Nanjing 210042, China

**Keywords:** three-dimensional bioprinting, vascularized channels, extrusion-based printing, drop-based printing, hydrogel, perfusion

## Abstract

Three-dimensional bioprinting has emerged as one of the manufacturing approaches that could potentially fabricate vascularized channels, which is helpful to culture tissues in vitro. In this paper, we report a novel approach to fabricate 3D perfusable channels by using the combination of extrusion and inkjet techniques in an integrated manufacture process. To achieve this, firstly we investigate the theoretical model to analyze influencing factors of structural dimensions of the printed parts like the printing speed, pressure, dispensing time, and voltage. In the experiment, photocurable hydrogel was printed to form a self-supporting structure with internal channel grooves. When the desired height of hydrogel was reached, the dual print-head was switched to the piezoelectric nozzle immediately, and the sacrificial material was printed by the changed nozzle on the printed hydrogel layer. Then, the extrusion nozzle was switched to print the next hydrogel layer. Once the printing of the internal construct was finished, hydrogel was extruded to wrap the entire structure, and the construct was immersed in a CaCl_2_ solution to crosslink. After that, the channel was formed by removing the sacrificial material. This approach can potentially provide a strategy for fabricating 3D vascularized channels and advance the development of culturing thick tissues in vitro.

## 1. Introduction

Nearly a million patients need transplants because of organ failure, but the number of transplants available is still far from enough. Biological tissue engineering technology provides a direction to solve the problem of human tissue organ transplantation. Three-dimensional (3D) printing can produce complex multi-material structures with controllable shape structures and controllable material components, which has become the first choice of tissue engineering [1,2,3]. A basic prerequisite for constructing thick tissue engineering (thickness > 200 μm) is the formation of an efficient 3D vascular exchange system, due to the limitation of extracellular matrix (ECM) penetration depth. Three-dimensional vascular system can promote the exchange of oxygen, nutrients, growth factors, and metabolic wastes between cells and the internal environment [4,5]. Thus, the development of a biomimetic 3D vascular system would be of vital importance [6,7].

There are three main strategies to create vessels in vitro in the field of bioprinting: extrusion-based, drop-based, and laser-based bioprinting [8]. Each technique accounts for a different range of material, manufacturing time, resolution, and design limitation [9,10].

Extrusion-based bioprinting, the most common technique, consists of dispensing a continuous filament through a nozzle using a piston, a screw, or pneumatic system to construct a 3D structure layer-by-layer. This technique provides an accurate ink deposition (resolution > 100 µm) and a homogenous cell distribution with faster printing speed. In addition, the bioink can reach an excellent structural integrity [11], and a wide range from low to high viscosity bioink, from 30 mPa·s to 6 × 10^7^ mPa·s [12,13], can be extruded with high mechanical strength. Kolesky et al. developed a water-soluble biomaterial composed of polysaccharides and PF127, which is biocompatible and can be printed and dissolved under mild conditions. The material can be printed in combination with cellular ink to produce heterogeneous cell structures with interconnected vascular networks [14]. Drop-based bioprinting is based on the ejection of a cell-laden bioink out of a nozzle in the form of droplets via piezoelectric, thermal, or electrostatic actuators to allow the drops to fall on a substrate with a precise control over the position using a non-contact approach [15]. The technique has higher resolution (resolution: 20–100 µm) than extrusion-based bioprinting but takes less time to print. Schöneberg et al. used drop-based bioprinting to generate blood vessel models in vitro mimicking a native vascular channel [16]. Their method was to deposit droplets of three different hydrogels containing three different cell types in a definite spatial layout to simulate the different layers of a native vascular channel. They succeeded to reproduce the three layers of a native vascular channel by reconstituting the tunica adventitia (matrix of fibroblast), tunica media (elastic SMCs), and the tunica intima (endothelium). Laser-assisted bioprinting (LAB) involves utilizing a laser pulse to separate cells from a donor slide, depositing them on a given substrate in a specific pattern [10]. The cell layer is deposited as evenly as possible under the donor slide. When a laser pulse is received from the top of the donor glass slide, bubbles form beneath the cell layer, causing the cells to detach. This technique combines the highest cell survival rate and the highest resolution but does not print efficiently [17]. Wu et al. used LAB to induce cell proliferation and maturation to form the vascular networks [18].

Due to the lack of universality of the above methods in the vascular networks, the construction of the 3D vascular networks within tissue engineering is limited. As a result, successful tissue engineering is limited to thin tissues such as the cornea, skin, and bladder. For large tissues (thickness > 200 μm), the lack of necessary 3D vascular system has seriously affected the progress of large tissue and organ engineering such as lung, liver, kidney, and pancreas [19,20]. In order to print 3D vascularized channels, the printer must have the ability to deposit at least two bioinks and have sufficient printing resolution. The bioink used must meet the requirements of both designed vascularized channels and 3D printing technology.

Here we propose a novel bioprinting approach on how to print sacrificial ink into the hydrogel matrix to fabricate perfusable channels, based on the combination of extrusion and inkjet techniques. In this work, a dual-head printer with a pneumatic extrusion nozzle and a piezoelectric nozzle was used. Hydrogel was printed layer-by-layer firstly and exposed to ultraviolet light with a wavelength of 405 nm, which formed a self-supporting matrix with the grooves of the internal channels. When the desired height of hydrogel was achieved, the piezoelectric nozzle was switched to eject sacrificial ink within the printed hydrogel layer to fill the internal grooves. Once the printing of the internal structure was finished, hydrogel was extruded to wrap the whole structure. Then, it was immersed in a CaCl_2_ solution to fully crosslink the structure, and the sacrificial ink was removed by water to form 3D hollow channels. Compared with other methods, hydrogel was pneumatically extruded to allow a self-supporting viscous structure and provide sufficient mechanical strength to ensure the formation of a 3D structure. The non-contact piezoelectric valve has the advantage of ejecting various sacrificial ink quantitatively and accurately [21,22] at a fixed point, which can eject in a narrow space as small as 100 μm to fabricate tunable channels within the hydrogel matrix. We report an important advance as our approach can create user-defined (shape, size, and location) and perfusable channels embedded within a hydrogel matrix in an integrated manufacture process.

## 2. Materials and Methods

### 2.1. Bioprinting System

The experiments were performed using a 3D multi-material bioprinter based on the SAM4E8E (Atmel, USA) processor. The printheads move across the xy horizontal plane while the molding platform only moves along the z-axis. Three extrusion nozzles and a piezoelectric nozzle were installed on the x-axis. The printing pressure of three extrusion nozzles can be independently adjusted using individual air pressure regulators. Hydrogel deposition in each extrusion nozzle was controlled by opening and closing the solenoid valve. One of the extrusion nozzles and a piezoelectric nozzle were used.

A computer-aided design (CAD) software SolidWorks (Dassault Systemes, Waltham, MA, USA) was utilized to create the 3D models for printing and generate the final stereolithography (STL) files. The slicing software Simplify 3D was utilized to create G-code. The printheads moved according to G-code instructions, depositing materials where they were initially programmed. Finally, the G-code was sent to the bioprinter by printing control software, which was also in charge of monitoring the printing process.

### 2.2. Materials Preparation

Ink formulations were prepared by dissolving sodium alginate (Aladdin, Shanghai, China) at different concentrations in deionized water in the presence of a photoinitiator 2-Hydroxy-4′-(2-hydroxyethoxy)-2-methylpropiophenone (Aladdin, Shanghai, China), polyethylene glycol diacrylate (PEGDA) (Aladdin, Shanghai, China). For instance, to prepare a 6% sodium alginate, 0.1 g of photoinitiator, and 3 g of PEGDA were dissolved in 100 mL of deionized water in a glass vial. Sodium alginate (6 g) was then added into the solution, and the solution was stirred overnight at room temperature. Sodium alginate has high mechanical strength and good biocompatibility without damaging the structural integrity. PEGDA can be gelled quickly at room temperature in the presence of a photoinitiator when exposed to ultraviolet light. Considering the previous reports [23], three different concentrations were used for sodium alginate (6%, 8%, and 10% (w/v)). The viscosity of hydrogel was tested by a rotational viscometer (Brookfield, MA, USA). Pluronic F-127 (PF127) was used as a sacrificial ink [24]. Sacrificial ink, PF127 (Sigma, Aldrich, Shanghai, China), was prepared by dissolving 30 g PF127 in 100 mL of deionized water dyed blue (30% (w/v)) at 4 °C overnight.

### 2.3. Parameters Optimization

Dots and lines are the minimum forming units for 3D printing and determine the forming size, while the characteristic size is closely related to the printing parameters and the properties of the material. As a result, the ideal concentration of hydrogel was firstly determined. Then, the influence of key printing parameters such as printing speed, extrusion pressure, voltage, and dispensing time on the molding effect was studied in the printable area to obtain the optimal combination of printing parameters. Images of the printed structures were captured by an optical microscope (Olympus CKX31, Tokyo, Japan).

### 2.4. 3D Printing of Sacrificial Ink Embedded within Hydrogel Matrix

Our approach utilized printing of a sacrificial ink PF127 within the freshly printed hydrogel layer to fabricate perfusable 3D channels by using the combination of extrusion and inkjet techniques in an integrated manufacture process. Our approach is summarized in Figure 1a. Briefly, the hydrogel was printed layer-by-layer firstly and exposed to ultraviolet light during the printing for simultaneous photopolymerization, which formed a self-supporting matrix with the grooves of the internal channels. When the desired matrix thickness was achieved, the dual-head printer was switched to piezoelectric nozzle, and the sacrificial ink was directly printed within this layer to fill the groove. This process was repeated as required to complete the printing process.

Specially, the extrusion nozzle (0.6 mm) and the piezoelectric nozzle (0.2 mm) were used, resulting in the slice path planning for the height and width of the hydrogel layer, which were 0.5 mm, and for the height and width of the PF127 layer, which were 0.25 mm, so two layers of PF127 were printed to match the thickness of a hydrogel layer to achieve contour molding in the same layer, as shown in Figure 1b.

As shown in Figure 1c, once the printing of the desired construct was completed, the structure continued to be exposed to ultraviolet light to cure for half an hour. Then, the structure was immersed in a CaCl_2_ solution to crosslink. After crosslinking, the structure was placed at 4 °C for one hour. Channels within the hydrogel matrix were then perfused with the dyed deionized water. Perfusion of the 3D channels was tested by using dyed deionized water, which was injected manually with a syringe.

## 3. Results and Discussion

### 3.1. Parameters Optimization

#### 3.1.1. Parameters Optimization of Hydrogel Ink

The viscosity of hydrogel ink was related to the proportion of hydrogel, including polymer solvent miscibility, polymer molecular weight, and polymer concentration, which determined the printing parameters. In addition, proportion of hydrogel also determined the material flow rate and printed line height. Almost all of the printing parameters have to be adjusted with the change of hydrogel proportion [25].

Tests were performed on glass slides to determine the optimal printing parameters such as concentration, the printing speed, and pressure. Firstly, the optimum concentration of sodium alginate was determined. A continuous s-shaped model was printed on a glass slide. The results for each concentration (6%, 8%, and 10% (w/v)) are shown in Figure 2. For 6% (w/v) ink, the hydrogel viscosity was significantly too low, and the structure collapsed. For 8% (w/v) ink, the printed filaments connected to form a seamless self-support base. The viscosity of hydrogel was tested to be 2600 mPa·s. For 10% (w/v) ink, the high viscosity and lack of fluidity led to the unconnected filaments and the formation of gaps.

Extruded hydrogel usually results in deformation from the initial shape as a consequence of standing their weight and slow gelation rates [26]. Furthermore, the filaments are never cylindrical, even if we use a cylinder-shaped nozzle. Hence, we decided to evaluate the resolution of printed filaments by width and height. Generally, the height of the filaments was determined by the diameter of the nozzles and the height of the nozzles from the platform [27]. We observed that the printing pressure and speed are critical factors for the width of the filaments. The printing pressure and speed are a pair of cooperative parameters. The ratio of them should be within a reasonable range in order to print a uniform filament. Line tests were performed three times on glass slides with an extrusion nozzle to determine the optimal printing parameters (pressure and speed). With the printing speed within 250–850 mm/min, the pressure ranged from 0.100 MPa to 0.175 MPa. In Figure 3, test results show the average width of the printed filaments under different combinations of pressure and speed.

For hydrogel ink, the width of the filaments significantly increased with increasing pressure at a fixed printing speed. For instance, the width increased from 783 μm, for 0.100 MPa (250 mm/min), to 1283 μm, for 0.125 MPa, to 2025 μm, for 0.150 MPa, and to 2680 μm, for 0.175 MPa. The width decreased with increasing speed. When filaments printed at 250 mm/min were compared with 400 mm/min, width decreased from 783 to 472 μm, for 0.100 MPa; 1283 to 1024 μm, for 0.125 MPa; 2025 to 1003 μm, for 0.150 MPa; and 2680 to 1133 μm, for 0.175 MPa. Test results for all of the hydrogel formulations are given in Table 1.

#### 3.1.2. Parameters Optimization of PF127 Ink

The sacrificial ink PF127 was printed by ejecting out of a piezoelectric nozzle in the form of droplets to allow the droplets to form a line. The main parameters for evaluating the connecting points into a line are the spaces between droplets T and droplets diameter D, and the droplets spacing should match the droplets diameter, as shown in Figure 4. If T > D, gaps form between points. If T ≤ D, overlaps form between points, forming surface wrinkles that affect the accuracy of the forming lines. In order to assess the quality of the printed filaments, the experiments were carried out by fixing other variables and studying the rule that one parameter affected the line formation of connecting points. The process test settings are shown in Table 2.

Additionally, droplets diameter D and spaces between droplets T measurements were obtained to characterize the effects of the critical printing parameters (dispensing time, the printing voltage, and speed) on the printed structure. The results of the measurements are summarized in Figure 5. These relationships between the printed structure size and the printing parameters can be applied to optimize the printing parameters. Figure 5a,b gives the droplets diameter D and spaces between droplets T with different printing speeds and dispensing time when the voltage percentage was 35%. Here, PF127 was printed at different speeds in 3 groups at a fixed dispensing time: 2.2, 2.4, 2.6 ms, respectively. Each group was printed three times, and the values of T and D at five points were measured to obtain the average value. The result reveals that the increasing speed leads to an increase in T. As can be seen from Figure 5a, when the speed was less than 550 mm/min, T was less than D (0.25mm), and the droplets began to overlap and connect to form a line. The effect of the printing speed and dispensing time on D is demonstrated in Figure 5b; it is seen that D increased with the increasing dispensing time and decreasing speed. Meanwhile, as indicated in Figure 5c,d, T and D were also sensitive to both of the dispensing time and voltage when the speed was 500mm/min. T decreased with the increasing dispensing time and decreasing voltage. Figure 5d reveals that D is positively dependent on the dispensing time and the voltage. When T is 55%–70% of D, a better formed surface can be obtained [28]. Through the parameters optimization experiment, the optimal parameter can be determined according to the set width D and the corresponding T to print the uniform line with the set width.

Figure 6 depicts the optical images of PF127 droplets obtained by printing on slides with different printing parameters. As seen in Figure 6a,b, the spaces between droplets decreased obviously when the voltage was fixed to 30% and the dispensing time increased from 2.4 to 2.5 ms. Figure 6c–f shows the spaces between droplets decreased with decreasing speed (from 1400 to 500 mm/min) when the dispensing time was fixed to 2.2 ms and voltage was fixed to 30%. As shown in Figure 6f, the spaces between droplets decreased when the speed decreased; the droplets connected into a line. A wider uniform and continuous line is shown in Figure 6g when the voltage and dispensing time increased to 35% and 2.6 ms and the printing speed was 500 mm/min. Compared to Figure 6g, the line in Figure 6h became nonuniform when the voltage increased to 40%. Obviously, the increase in voltage results in an increase in width.

### 3.2. 3D Bioprinting of PF127 within Hydrogel Matrix to Fabricate Perfusable Channels

Three-dimensional digital designs and sizes are shown in Figure 7. Three-dimensional channels were within a hydrogel matrix (20 mm × 20 mm × 8 mm) with strut sizes of the channels in x-direction, y-direction, and z-direction of 9 mm × 3 mm × 2 mm, 3 mm × 3 mm × 4 mm, and 9 mm × 3 mm × 2 mm. In order to achieve the set height and width of the hydrogel and PF127, according to the parameters optimization results, the parameters were determined as follows. The printing pressure of the extrusion nozzle was set to 0.125 MPa, and the printing speed of extrusion nozzle was set to 550 mm/min. The printing speed, voltage percentage, and dispensing time of the piezoelectric nozzle were set to 500 mm/min, 30%, and 2.6 ms. The time of the total printing process was 36.04 min.

Figure 8 shows the corresponding printed structure. To reveal the 3D structure of the channels, blue paints were added in the sacrificial ink before printing. The overall view confirmed its structure in the 3D space, as shown in Figure 8a–c. It can be observed that the construct possessed a distinct 3D structure.

In this experiment, the optimal concentration of hydrogel (6%, 8%, and 10% (w/v)) was firstly determined and we decided to use the concentration of 8% (w/v). Then, we investigated the optimal printing parameters of hydrogel by conducting line tests on glass slides. The main purpose of these tests was to adjust the parameters such as the printing speed and pressure to obtain continuous and uniform lines. A wide range of printing speeds and pressures were tested to fabricate hydrogel filaments from 472 up to 2680 μm (Table 1). As expected, an increase in width with decreasing speed at a constant pressure and an increase in width with increasing pressure at a constant speed were obtained. As a result, 8% (w/v) hydrogel was printed at a print speed of 500 mm/min and a pressure of 0.300 MPa. For PF127, line tests were performed to determine the relationship between the printing parameters and line size. PF127 was printed by ejecting in the form of droplets via piezoelectric nozzle to allow the drops to form a line. To form a continuous and uniform line, PF127 was tested under different combinations of voltage, dispensing time, and speed. As a result, the PF127 line size was nearly equal to the groove size when PF127 was printed within the matrix to fill the grooves. Finally, we successfully fabricated the 3D channels within hydrogel matrix.

### 3.3. The Morphology of the Construct and Perfusion

The perfusion results are shown in Figure 9. In Figure 9a, the channel (in y-plane) is partially perfused. In Figure 9b, the two connected channels (in y-, z-planes) are perfused. In Figure 9c, all channels with 3D structure (in x-, y-, z-planes) within matrix are perfused.

As shown in Figure 10a, the section view of the channel reveals that the molding generated a real channel after washing away PF127. A thin sheet of hydrogel matrix containing the channel was cut off and observed with an optical microscope. In Figure 10b, a clear hollow circular section is shown. The shape of the hole slightly deviates from the rectangle, which was supposed to be affected by washing and slicing. The proposed construct possessed a 3D structure while having the hollow channels to perfuse. This is particularly important because the thin blood networks have a limited significance for the thick tissue culture in vitro.

## 4. Conclusions

In conclusion, we propose a method for the fabrication of 3D perfusable channels within photocurable hydrogels, specifically created by the combination of extrusion and inkjet techniques in an integrated manufacture process. The described method creates a new approach towards the fabrication of 3D vascularized channels to advance the development of culturing thick tissues in vitro. Using this method, it was possible to create a construct that is stable enough to support the structure as well as the 3D perfusable channels embedded within the hydrogel. Thus, it holds enormous potential for the fabrication of 3D vascularized channels of different shapes within the matrix to maintain the culture of thick tissues. However, much work is still required in this field, for example, resolution and shape fidelity.

## Figures and Tables

**Figure 1 micromachines-11-00237-f001:**
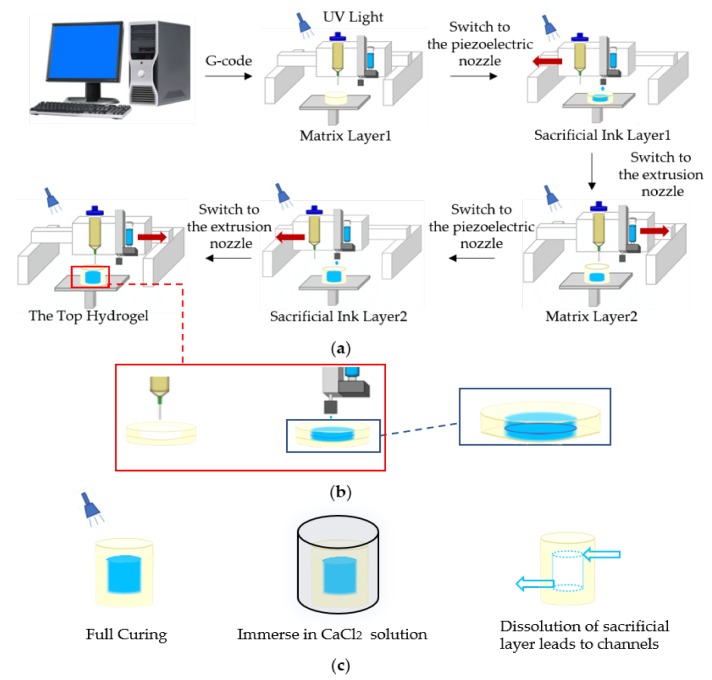
Schematic showing the printing approach to fabricate 3D channels embedded within hydrogel matrix: (**a**) sequential printing of photocurable matrix hydrogels with groove structure and a sacrificial ink within printed matrix layer-by-layer; (**b**) an enlarged image of a single layer, notably, because the diameter of extrusion nozzle and piezoelectric nozzle is different, a hydrogel layer requires two or three layers of sacrificial ink to match the thickness; (**c**) post-printing process including full curing, immersing in a CaCl_2_ solution to crosslink and remove sacrificial ink to fabricate channels.

**Figure 2 micromachines-11-00237-f002:**
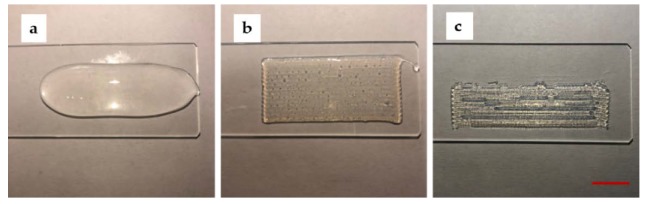
The printed results for concentration of sodium alginate are 6%, 8%, and 10% (w/v) from left to right (Scale bar: 1 cm). (**a**) the printed result for 6% (w/v) sodium alginate; (**b**) the printed result for 8% (w/v) sodium alginate; (**c**) the printed result for 10% (w/v) sodium alginate (Scale bar: 1 cm).

**Figure 3 micromachines-11-00237-f003:**
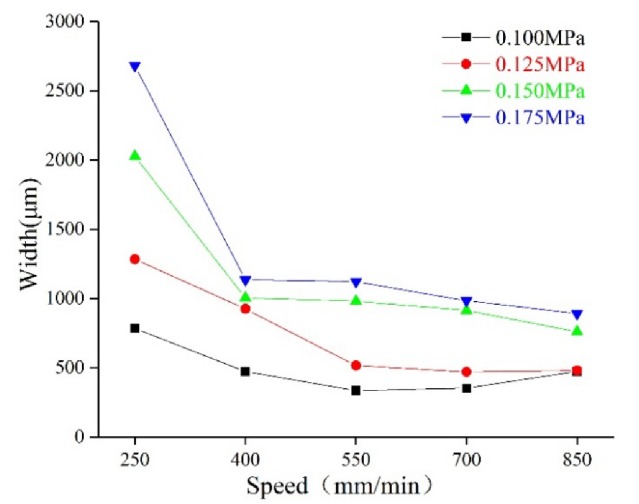
The molding relationship: dependence of the combination of extrusion pressure and speed on filament width.

**Figure 4 micromachines-11-00237-f004:**
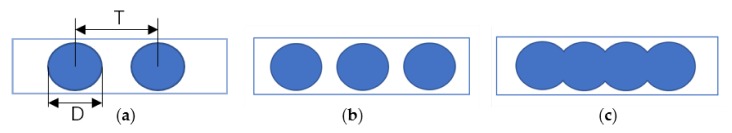
Schematic for connecting droplets into a line: (**a**) the printed droplets of diameter D and space between droplets T; (**b**) when T > D, there are gaps between the droplets; (**c**) when T ≤ D, droplets overlap to form a line.

**Figure 5 micromachines-11-00237-f005:**
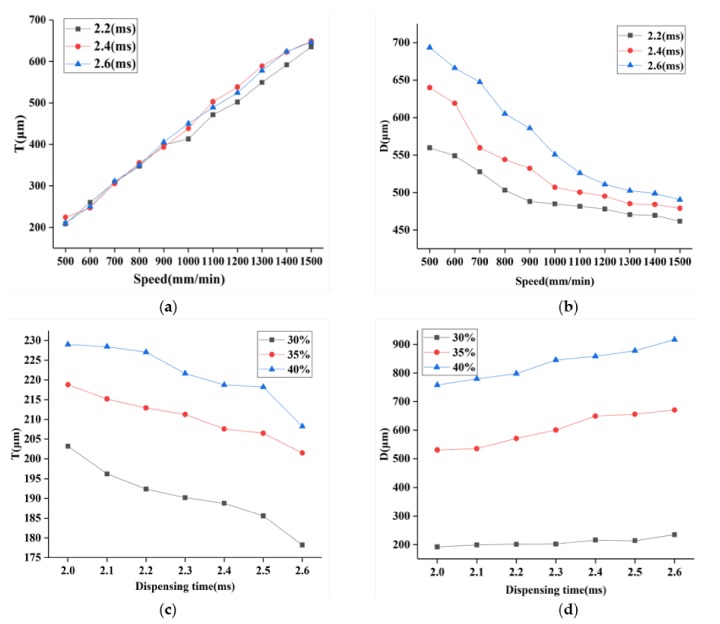
Characterization of the printed droplets and lines. (**a**) Mean spaces of the printed droplets corresponding to the speed at different dispensing time; (**b**) mean diameters of the printed droplets corresponding to the speed at different dispensing time; (**c**) mean spaces of the printed droplets corresponding to the dispensing time under different voltage; (**d**) mean diameters of the printed droplets corresponding to the dispensing time under different voltage.

**Figure 6 micromachines-11-00237-f006:**
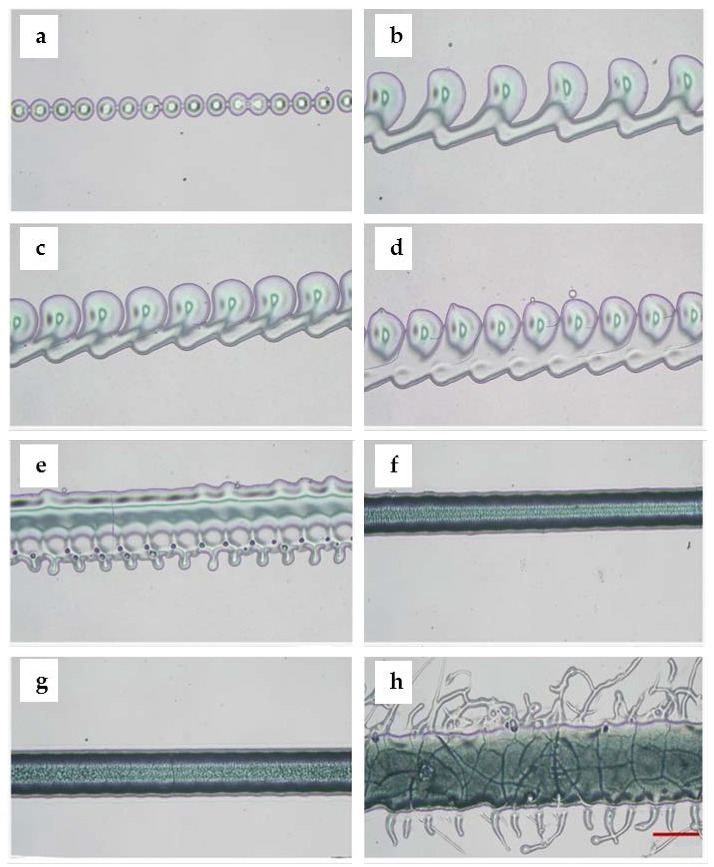
Optical microscope images of the printed droplets and lines with different parameters. (**a**) Droplets formed the continuous points when voltage was set to 30% and dispensing time was set to 2.4 ms; (**b**) droplets formed the connected points when voltage was set to 30%, dispensing time increased to 2.5 ms; (**c**) the spaces between droplets became smaller when voltage was fixed to 30%, dispensing time was fixed to 2.2 ms, but the printing speed was set to 1400 mm/min; (**d**) connected droplets formed a filament when the printing speed was set to 1000 mm/min; (**e**) droplets formed the uneven filament when the printing speed decreased to 900 mm/min; (**f**) droplets formed the uniform filament when the printing speed was set to 500 mm/min; (**g**) droplets formed a more wide uniform filament when voltage was changed to 35%, dispensing time was set to 2.6 ms, and the printing speed was set to 500 mm/min; (**h**) the filament became wider and uneven when the voltage increased to 40% (scale bar: 400 μm).

**Figure 7 micromachines-11-00237-f007:**
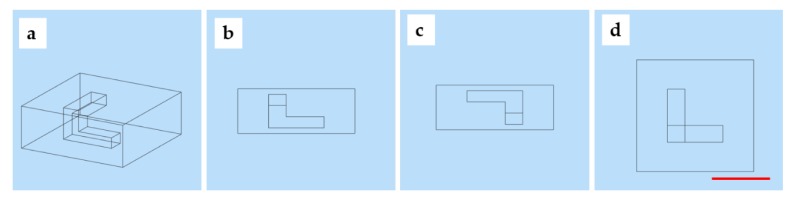
The designed model: (**a**) overall design; (**b**) front view; (**c**) left view; (**d**) top view (scale bar: 1 cm).

**Figure 8 micromachines-11-00237-f008:**
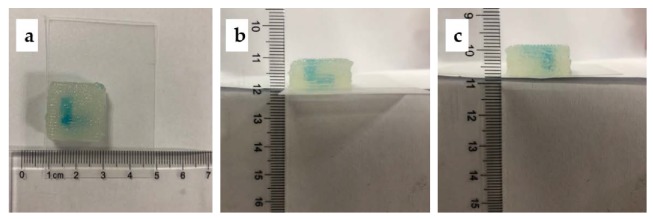
The corresponding printed constructs: (**a**) the top view of the printed construct; (**b**) the front view of the printed construct; (**c**) the left view of the printed construct.

**Figure 9 micromachines-11-00237-f009:**
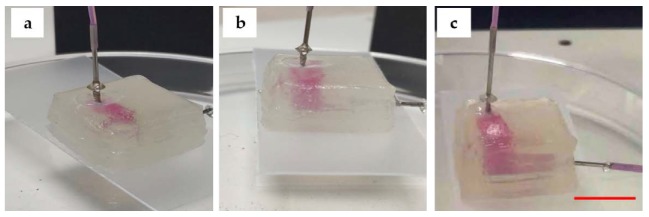
Perfusion: (**a**) the channel is partially perfused; (**b**) the two connected channels after the perfusion; (**c**) all channels with 3D structure were perfused with pink deionized water (Scale bar: 1 cm).

**Figure 10 micromachines-11-00237-f010:**
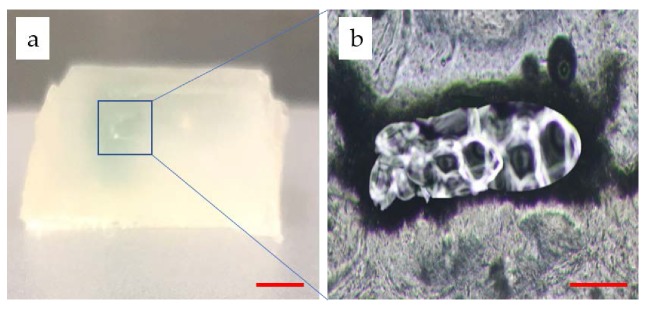
Morphology of the channels: (**a**) the section view of the hydrogel matrix (Scale bar: 4 mm); (**b**) the section view of a thin sheet of hydrogel matrix containing the channel (Scale bar: 400 μm).

**Table 1 micromachines-11-00237-t001:** The experimental parameters of matrix hydrogels.

Pressure (MPa)	Speed (mm/min)	Width (μm)
0.100	250–850	783–472
0.125	250–850	1283–479
0.150	250–850	2025–759
0.175	250–850	2680–890

**Table 2 micromachines-11-00237-t002:** The experimental parameters of sacrificial ink.

Dispensing Time (ms)	Voltage Percentage (%)	Speed (mm/min)	T (μm)	D (μm)
2.2	35	500–1500	208.73–635.4	490.47–693.47
2.4	35	500–1500	224.25–648.99	479–639.95
2.6	35	500–1500	210.45–646.37	461.63–559.67
2.0–2.6	30	500	178.21–203.19	192.11–234.70
2.0–2.6	35	500	201.47–218.89	530.53–670.74
2.0–2.6	40	500	208.23–229	758.32–917.2

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
