# Peer review of "The Research on Multi-Material 3D Vascularized Network Integrated Printing Technology"

_micromachines, 2020, doi:10.3390/mi11030237_

Round 1

Reviewer 1 Report

The manuscript submitted by Yang et.al described an integrated fabrication method to manufacture a 3D vascularized channel by combining an extrusion-based printer with an inkjet-based one. Firstly they optimized different printing settings including speed, pressure for hydrogel extrusion printer, dispensing time and voltage for PF-127 inkjet printing. Then they demonstrated the fabrication of a 3D channel inside the hydrogel by this hyphened printing technology. The idea of combing extrusion with inkjet-based printing technology is interesting, but this manuscript still has a lot of parts needed to be improved, so a major revision will be recommended.

Major issues:

The integration of different 3D printing technologies in a single process for various purposes (eg. Increasing functionalities, spatial complexity) is not rare currently, but I am not fully convinced by the significance of the integrated technology presented in this manuscript, I hope the authors could explain it more. More specifically, have the author tried to print the same structure with only extrusion printing hydrogel without sacrificial material? Given the channel dimension is not small here, I am wondering if there is a need of sacrificial material. If we do need it, it is always helpful to have a comparison here to show your integrated printing technology is superior. I understand that the inkjet-based printer has a better resolution than that of the extrusion-based one, but the overall resolution is always limited by the lower one, eg the smallest channel dimension in this manuscript is limited by the extrusion hydrogel printer. Is it easier or more rational to integrate multiple printheads with same printing technology like extrusion or inkjet-based?

Minor issues:

Line 56, page 2. ‘higher resolution (20-100nm)..’, should it be µm? currently only 2PP can achieve nm resolution. line 108, page 3. I am assuming the ALADDIN is the chemical company? the location of the company also needs to be listed, it should be (Aladdin, XXX), same for sigma. For the fabrication process, is the sacrificial completely sealed and wrapped by hydrogel? If so, how did the author remove it? It would be quite useful for the readers to understand if the author can add a short video about the printing process as a supporting information. Also what is the wavelength of the UV light? What is the initiator you used, they are all very important information. How long did you polymerize for each hydrogel layer? In figure 2 b), a lot of bubbles can be observed, maybe the author can try sonication or vacuum degassing to remove the bubbles that can affect the printing. what is the slicing layer thickness of the extrusion and inkjet printer? What is the printing time? They should be listed. in figure 3, the unit of Y axis, width (mm), should it be µm? for the graph type, it should be scatter with STRAIGHT line, not with smooth line, please see the figure 6, which is right. in Figure 6, what voltage was applied for a), b)? what speed was used for c), d)? they should be listed. Why figure b) started from 1000, not from 500, it should be consistent with a)? in b), blue line, why the D increases when raising speed from 1000 to 1100, then decreases when speed raising further? Same question for the others in b) and c). How many repeats the author performed for figure 3, and 6, what is the standard deviation (error bar) of each measurement? How do you know the fluctuation of the data is not caused by manual error? there is a lack of several important information in the parameters optimization parts for both extrusion and inkjet printing, what the optimized settings the author chosed? Based on what? Line 241, page 8, 0.285-0.300 MPa. I have not found any information about this pressure in part 3.1.1 optimisation part and figure 3. Why did you chose this pressure even you did not try it in optimization part? Or you did not list it? in figure 10, the quality of a) should be improved. For b), I can see there are some crosslinked structure in the channel, so the channel is not as the author claimed as hollow in the manuscript. Do the authors know what that structure is? Uncleaned and polymerized PF-127 or excess of hydrogel?

Reviewer 2 Report

Title: The Research on Multi-material 3D Vascularized Network Integrated Printing Technology

Type: Research article

Authors have proposed a novel bioprinting approach for the fabrication of 3D perfusable channels within photocurable hydrogels using the combination of extrusion and inkjet technique in an integrated manufacture process. Authors have carried out experiments to prove its fidelity, however, in depth explanations are not addressed appropriately in the results and discussion section. Authors should correlate their work with the reported papers and should justify their claims. Moreover, lack of data has also been noticed. In order to claim this model as a potential candidate for maintaining the culture of thick tissue laminates, there should be some preliminary data to support it, which seems to be lacking. Additionally, there are lots of casual mistakes in terms of using representation or formatting errors and confusing statements that should be taken seriously and hence, English correction is recommended. Authors should proof read the manuscript thoroughly and carefully before re-submitting the paper. After reviewing it carefully, authors are noted with the following comments:

Comments

Use of abbreviations should be consistent throughout the manuscript. For instance, use of CaCl2. It is differently represented in section 2.4-line 136 and figure 1 (c). Similarly, with the term layer-by-layer. It is again differently mentioned in section 2.4-line 131 and line 144. Another example is in section 2.2-line 111, use of MeAlg without its full form. Additionally, the authors should be careful and consistent while using the units. For instance use of µm in line 36 and in line 46. Another example is use of either kPa, which is differently designated in the same statement itself in section 3.1.1-line 177. Authors should also be noted the use of either MPa or kPa. Authors should follow a single and appropriate way of representation. It should be consistent throughout the manuscript. Kindly note, that these are few examples. There are many in the entire manuscript.

Section 2.2, line 117-119: Authors should rephrase the statement appropriately and also be consistent with the use of PF127. Once abbreviated, it should be same throughout the manuscript. For instance, in the same statement, use of Pluronic is in different way: Pluronic F-127 and again at the end of the same line it is used as Pluronic F127. Consistency should be maintained throughout the manuscript.

Section 2.3, line 126: Athurs should provide the model number for the microscope used.

Section 3.1.1, line 155-157: Authors talked about hydrogel viscosity, but nowhere in the manuscript is rheological data seen. On what context authors can talk about significant difference in hydrogel viscosity without any experimental rheological data? Please explain. Authors should provide quantified data for making such claims.

Section 3.1.1, line 161-162: It is very difficult to understand what the authors have tried to deliver through this statement. Authors should reframe this sentence.

Authors should mention in the material and methods about the nozzle diameter that was used for printing. Section 3.1.1, line 175-181: It is difficult to understand the results without knowing the actual nozzle diameter.

Figure 3, on x-axis, what is the correct represented unit mm or µm? Because in the text it is mentioned as µm. Consistency should be maintained throughout the manuscript.

Figure 4, line 13: What does the word “evening” mean in this context?

Section 3.1.2, line 198-205: Authors should analyze the results again and re-write it appropriately in the manuscript as it seems very confusing to read. Additionally, figure numbers are represented incorrectly in the text adding to more confusion. For instance, this explanation is supposed to be for figure 5. However, authors mentioned it as figure 7 and figure 8 for the same explanation. Authors should take these points very seriously and correct it carefully.

Section 3.2, line 242-243: Why did authors chose voltage percentage and speed as 30% and 550 mm/min for piezoelectric nozzle, whereas in section 3.1.2, line 203-204, it is mentioned that a uniform and continuous line is obtained with voltage 35% at a speed of 500 mm/min? Please explain.

Section 3.2, line 243-248: The process of printing is already mentioned in section 2.4. Hence, it is not required. Also the diameter of the nozzles (line 249) should be mentioned previously in materials and methods for the readers to understand the results obtained in section 3.1.1. Authors should avoid this kind of redundancy to make the results more focused.

Section 3.3, line 278-281: Again, it should be mentioned in materials and methods. This section is to dictate about the results only. Authors should avoid this kind of redundancy to make the results more focused.

Figure 9 (c), line 293: Should not it be x-, y-, z-planes? All three of them as mentioned in the text (Section 3.3, line 284). Authors should check and correct it.

Round 2

Reviewer 1 Report

I am ok with the response from the authors, there is only one minor issue in point 7 'waht is the printing time?', i meant to ask how long it took to complete the print.

Author Response

Dear reviewer,

Please see the attachment.Thank you.

Kind regards,

Shuai Yang

Reviewer 2 Report

Authors have proposed a novel bioprinting approach for the fabrication of 3D perfusable channels within photocurable hydrogels using the combination of extrusion and inkjet technique in an integrated manufacture process. Authors have addressed to most of the comments and corrected it. However, there are still lots of grammatical and casual mistakes in terms of using representation or formatting errors and few confusing statements that should be taken seriously. Hence, English correction is recommended. Authors are advised to proof read the manuscript thoroughly and carefully before re-submitting the paper again. After reviewing it carefully, authors are noted with the following comments:

Comments

  • This manuscript is specific to providing a new strategy for fabricating 3D perfusable channels. Authors have carried out experiments and reported the results to prove its fidelity claiming this model as a potential candidate for maintaining the culture of thick tissue laminates. Using the term “bioink” seems inappropriate and misleading as it stands for “hydrogel mixed with cells”. In this manuscript cells are not mixed with the prepared hydrogel and also there are no cell culture data. Authors should correct it or justify using the term “bioink”.
  • Use of abbreviations should be consistent throughout the manuscript. For instance, use of 3D (three-dimensional). It is already abbreviated in the beginning, hence, it should be followed consistently throughout. Another example is the use of units in section 1, line 36, the use of “200µm”. Also in line 90, use of “micron”. Although authors have done the correction at some places but avoided at other places. It should be represented in the same manner throughout the manuscript. Similarly, with the term “layer-by-layer”. It is again differently mentioned at different places. For instance, Figure 1, line 164 and section 2.4, line 140. Again, the use of the word table. It should either be “Table” or “table”. For instance, section 3.1.1, line 225 and section 3.1.2, line 235. There are also spelling mistakes. For instance, section 3.1.2, line 296 “increased”. Authors should proof read the manuscript thoroughly. Casual mistakes are not acceptable. Kindly note, that these are few examples. There are still many in the entire manuscript.
  • Section 3.1.1, line 186-187: This section is meant to dictate only the result. The instrument used for determining rheological properties should be mentioned in materials and methods and the result should only be mentioned here. Authors should provide the rheological data for supporting their reported value.
  • Section 3.2, line 397-398: “The experiment was ….. afternoon”. Does it have any relevance with the reported data? If so then authors should explain. If not then this information is not required.

Author Response

Dear Reviewer,

Please see the attachment.Thank you.

Kind Regards,

Shuai Yang

Round 3

Reviewer 2 Report

The authors have adequately addressed my concerns. There are still grammatical errors and the writing is still a bit cumbersome but that can be corrected by the editing staff.